# Carbamazepine for Chronic Muscle Pain: A Retrospective Assessment of Indications, Side Effects, and Treatment Response

**DOI:** 10.3390/brainsci13010123

**Published:** 2023-01-10

**Authors:** Tabea M. Dyong, Burkhard Gess, Christina Dumke, Roman Rolke, Maike F. Dohrn

**Affiliations:** 1Department of Neurology, Medical Faculty, RWTH Aachen University, 52074 Aachen, Germany; 2Department of Palliative Medicine, Medical Faculty, RWTH Aachen University, 52074 Aachen, Germany; 3Dr. John T. Macdonald Foundation, Department of Human Genetics, John P. Hussman Institute for Human Genomics, Miller School of Medicine, University of Miami, Miami, FL 33136, USA

**Keywords:** carbamazepine, myalgia, myopathy, quantitative sensory testing, neuropathic pain

## Abstract

Myopathies fall under the umbrella of rare diseases, however, muscle pain is a relevant, under-recognized symptom with limited treatment options. Carbamazepine is an oral sodium channel blocker approved for the treatment of seizures and neuropathic pain. In 54 individuals receiving carbamazepine for muscle pain, we retrospectively assessed the subjective treatment response, side effects, and reasons for carbamazepine discontinuation. The underlying diagnoses leading to muscle pain were diverse, ranging from metabolic (*n* = 5) and other hereditary (*n* = 9) to acquired (*n* = 2) myopathies and myotonia syndromes (*n* = 22). Under carbamazepine (daily dose 254 ± 138 mg), patients reported a significant reduction of pain, quantified by an 11-point numeric rating scale (−1.9 ± 1.8, *p* < 0.001). Compared to age- and sex-matched controls, our sensory assessment revealed a significant dysfunction of Aδ-nerve fibers in patients with chronic muscle pain. Neuropathic pain components identified by the painDETECT questionnaire or quantitative sensory testing did not seem to influence the reported treatment response. Side effects (*n* = 18) such as fatigue, elevated liver enzymes, and diarrhea, as well as lack of pain improvement (*n* = 6), led to carbamazepine discontinuation in 44.4% (24/54). Mediated by dysfunctional Aδ-nerve fibers, muscle pain is common in a variety of myopathies. Carbamazepine may reduce pain levels, but comes with therapy-limiting side effects.

## 1. Introduction

Carbamazepine is an approved treatment option for neuropathic pain, e.g., in diabetic neuropathies [1] or trigeminal neuralgia [2,3]. Muscle disorders are rare diseases with heterogeneous etiology, severity, and symptom patterns [4]. For most patients, there are neither curative nor specific symptomatic medications available.

Muscle pain is a common symptom experienced by patients with neuromuscular diseases [5,6,7] that, despite its impact on quality of life [8,9,10,11], has not been sufficiently addressed to date [5]. According to multiple databases, no significant progress has been made in the past five years toward the identification or approval of symptomatic drugs for muscle pain. The exception is mexiletine, an antiarrhythmic sodium channel blocker that has recently been approved for non-dystrophic myotonia [12]. With its risk of cardiac arrhythmia, however, mexiletine cannot be prescribed to other myotonia patients, therefore leaving the most frequent forms untreated. Besides myotonia [13], several studies describe the presence and relevance of muscle pain in diagnoses such as myositis [14], facio-scapulo-humeral muscle dystrophy (FSHD) [15], other muscular dystrophy syndromes [16], McArdle disease [17], and mitochondrial myopathies [18].

Like mexiletine, carbamazepine blocks voltage-gated sodium channels that are mainly expressed in peripheral C and Aδ nerve fibers [4]. Dysfunction of these small, un- or thinly myelinated fibers is associated with several chronic pain disorders, including small fiber neuropathies and fibromyalgia [19,20,21,22]. In contrast to neuropathic pain, muscle pain is considered to be nociceptive. Its localization is not superficial, but deep, its character pounding, burning, or cramp-like [4,23,24]. Recent studies show that there is a difference in experienced muscle pain in men and women [25]. The underlying mechanisms of muscle pain are not fully understood.

In this study, we screened a cohort of 560 myopathy patients for reported muscle pain. In 54 of these, carbamazepine had been prescribed to control chronic muscle pain within the past four years. We describe the etiological heterogeneity of muscle pain, characterize these patients, and assess both treatment response [26] and side effects of carbamazepine. In a subset of 24 out of 54 myopathy patients, we performed quantitative sensory testing (QST) [27,28,29,30,31] to further explore the complete somatosensory phenotype, including tests for small and large nerve fiber function. This approach allowed us to exclude or characterize an accompanying neuropathic pain syndrome and propose a new pathomechanistic hypothesis to explain the effects of voltage-gated sodium channel blockers in myopathy.

## 2. Materials and Methods

### 2.1. Inclusion and Exclusion Criteria

We enrolled adult patients (>18 years) with chronic muscle pain that had been diagnosed with a hereditary or acquired muscle disease and received at least one prescription of carbamazepine as a symptomatic pain medication. Patients with carbamazepine treatment due to epilepsy, trigeminal neuralgia, or diabetic neuropathy were excluded, as were patients that were treated with oxcarbazepine. The underlying muscle disease had been diagnosed following standard procedures, including clinical examinations (Appendix A), electromyography, muscle MRI, biopsies, or genetic testing. All patients gave written informed consent.

### 2.2. Patient Selection

We screened clinical records belonging to 1683 patients overall and 3168 visits from 2016 to 2018. Out of these, 560 patients had an underlying myopathy and 54 of them were treated with carbamazepine (Figure 1a,b). Twenty-four patients agreed to come to the center for a study visit, and thirty additional individuals answered our questionnaire via mail.

All clinical examinations were conducted by the same, trained personnel (TD, MFD) at the Neuromuscular Outpatient Clinic, Department of Neurology of the RWTH Aachen University Hospital, Aachen, Germany. We additionally evaluated records from previous visits. The study complied with the Declaration of Helsinki and was approved by the local ethics committee (EK00418).

### 2.3. Clinical Interview

Within the frame of a standardized clinical interview, we assessed the patients’ history including symptoms, medical and family history, as well as previous and current medications, focusing on pain characteristics, influencing factors, localization, dynamics, radiation, and intensity. Additionally, we assessed potential sources of bias including the intake of statins [32,33,34] or alcohol [35].

### 2.4. Screening for a Neuropathic Pain Component

To assess the pain type and character, we used the painDETECT questionnaire, which is designed to distinguish between nociceptive and neuropathic pain. With an overall score of fewer than 13 points, a neuropathic pain component is unlikely (<15%), whereas a score higher than 18 points indicates the presence of neuropathic with a probability above 90%. The overall sensitivity and specificity of the painDETECT questionnaire account for 84%, each [36].

### 2.5. Quantitative Sensory Testing (QST)

Detailed sensory phenotypes were evaluated by quantitative sensory testing (QST) in order to (1) identify the most prominently involved nerve fibers correlating with muscle pain and (2) draw conclusions on potentially co-existent neuropathy. The QST uses calibrated stimuli on defined skin areas. As a test area, we chose the right foot in 22 patients, the left foot in 1 patient, and the right hand in 1 patient. In our protocol, we used the standardized test battery and reference values from the German Research Network on Neuropathic Pain (DFNS) [27,37], consisting of 7 tests and 13 parameters [27,28,38]. This QST protocol allows testing for all relevant thermal and mechanical detection and pain thresholds across superficial and deep tissues without any a priori hypothesis. Each patient’s data were compared with age-, sex-, and area-matched reference values from our in-house database [27,28,30,31]. Most of the values were normally distributed after logarithmic transformation with the exception of paradoxical heat sensations (PHS), cold pain threshold (CPT), heat pain threshold (HPT), and vibration detection threshold (VDT) [28,30]. QST data were z-transformed in order to normalize the test results of individual patients: Z = (value_patient_ − mean_control_)/SD_control_. This enabled the distinction between loss and gain of fiber function [27,37]. For each parameter in both groups, the mean of the z-scores was calculated and compared by one-way ANOVA.

Pathological values in cold detection threshold (CDT), cold pain threshold (CPT), heat pain threshold (HPT), mechanical pain threshold (MPT), PHS, or pressure pain threshold (PPT) suggest malfunctioning of Aδ fibers whereas differences in HPT, warmth detection threshold (WDT), and also PPT, or CPT show a C fiber dysfunction. Deviations in mechanical detection threshold (MDT) and vibration detection threshold (VDT) point towards the involvement of Aβ fibers. Central sensitization is assumed if there are deviations in dynamic mechanical allodynia (DMA), HPT, or mechanical pain sensitivity (MPS). A positive wind-up ratio (WUR) suggests temporal summation [27,37,39].

### 2.6. Laboratory Blood Tests

Laboratory analyses were performed in peripheral blood in order to exclude potentially interfering causes of peripheral nerve dysfunction, for example, alcoholism, or diabetes mellitus, as well as to assess carbamazepine side effects.

### 2.7. Statistics

We imported the original dataset into GraphPad Prism 5. We tested Gaussian distribution using the D’Agostino and Pearson omnibus, Kolmogorow–Smirnow, and Shapiro–Wilk normality tests. To compare two subgroups within our study, we used the Student’s *t*-test if data followed a Gaussian distribution. Otherwise, we used the Mann–Whitney U test. To compare the same subgroup before and under carbamazepine treatment, we used the paired/dependent t-test. For group comparisons, we used one-way ANOVA or the Kruskal–Wallis test if non-parametric. The level of significance was defined as *p* < 0.05. For α-error correction, we used the Dunn’s test. Correlation analyses were performed with the Pearson or Spearman test.

## 3. Results

### 3.1. Patient Description

Our study represents a real-life cohort of individuals with muscle pain examined at a specialized rare disease and neuromuscular center (Figure 1a). Muscle pain was always related to a previously diagnosed hereditary or acquired myopathy, whereas fibromyalgia was not in focus in this work. Out of 54 patients (42 males and 12 females, mean age at examination: 50.9 ± 12.2 years, range 21–80 years), 46% suffered from hereditary muscle diseases, and 54% were diagnosed with an acquired myopathy (Figure 1b, Table 1).

Muscle diseases were assigned as myopathy of unknown origin (30%), benign cramp-fasciculation syndrome (20%), (suspected) neuromyotonia (14%), mitochondrial myopathy (6%), myofibrillar myopathy (4%), central core myopathy (4%), paramyotonia congenita (4%), FSH dystrophy type I (4%), centronuclear myopathy (4%), metabolic myopathy (2%), myotonia congenita type Becker (2%), phosphorylase deficiency type McArdle (2%), autoimmune necrotizing myopathy (2%), proximal myotonic dystrophy type II (2%), nemaline myopathy (2%), and myositic overlap syndrome (2%).

Independent of the underlying muscle pathology, myalgia was reported as the first symptom in 89% of the patients, followed by muscle cramps in 46%, fasciculations in 31%, muscle weakness in 22%, muscle stiffness in 11%, and atrophies in 2% (Table 1).

In 91% of the patients, no special incident was described prior to symptom onset. In the remaining 9%, the following triggers were subjectively attributed to the manifestation of muscle pain (2%, each): mononucleosis, hepatitis C, statin intake, physical stress/ sleep deprivation, and a fall. Patients described the disease course to be progressive (72%), stable (22%), or fluctuating (2%). Assessing the current symptoms, myalgia was the most common one found in 85%, followed by muscle weakness and cramps in 35%, fasciculations in 22%, muscle stiffness in 6%, joint pain in 4%, and hyperflexible joints, atrophies, headaches, and recurrent subluxations in 2% each (Table 1).

We obtained 90 CK values from 50 patients. Of these, 26 patients had an elevated CK level (>190 U/L in males and >170 U/L in females) at least once and 24 patients were within the normal range at all times. CK levels were most elevated in three patients with McArdle disease (2639 U/L), proximal myotonic dystrophy type II (2888 U/L), and centronuclear myopathy (1054 U/L). Previous rhabdomyolyses were reported by one patient with myopathy of unknown origin (Appendix A).

In 78%, the family history was not informative regarding neuromuscular disorders. Six percent reported first-degree relatives with comparable symptoms, although not being systematically assessed by a specialized physician. There was a positive family history for polyneuropathies, rheumatism, and Parkinson’s disease in 4% and for myotubular myopathies, amyotrophic lateral sclerosis, or spinal muscular atrophy (SMA) in 2%. Other diseases in the family history were gouty arthritis and fibromyalgia.

### 3.2. Pain Assessment

On average, the study participants had already been suffering from muscle pain for 14.2 ± 14.5 years (mean, SD) at the time of examination. The mean NRS during the study visit was 4.2 ± 2.7 (range 0–9, 10 in total). The pain dynamics were described as pain attacks (50%), persistent pain with pain attacks (38%), and persistent pain with slight fluctuations (13%). In addition to these pain qualities, 63% suffered from muscle cramps. Whole body pain was reported in 43%, followed by more localized manifestations at the limbs, including upper and lower leg muscles evenly in 39%, only the lower limbs in 13%, and 6% did not specify the localization of pain. Forty-three percent described their muscle pain to be generalized, and 6% did not clarify the exact localization. To describe the character of pain, the following attributes were most commonly used: soreness (62%), burning (21%), pricking (21%), pulling (21%), pressing (7%), cramping (7%), piercing (6%), heat-like (3%), and pulsing (3%). Pain radiation was reported in 14 patients (58%). A potential trigger of pain exacerbation was physical stress (57%). One patient reported an intensification of pain in the evening, while another patient experienced pain peaks in the morning. As mitigating factors, study participants reported warmth (15%) and physiotherapy (4%) including relaxation techniques, massages, and acupuncture, whereas others described moderate physical activity to be soothing. Fourteen patients (26%) had limited their professional situation either by changing from a mostly standing to a seated activity (2%) or by early retirement (24%).

In the painDETECT questionnaire completed by 44%, the mean score (±SD) was 7.6 ± 6.6 (35 in total) points. A neuropathic pain pattern (≥19 points) was found in two patients (8%), whereas three were classified in the borderline range (13–18 points). More than 80% of our cohort did not show any neuropathic pain component (≤12 points). For each designated neuropathic pain item, the majority (≥75%, *n* ≥ 18) of patients rated the frequency of occurrence as “never”, “hardly noticed”, “slightly”, or “moderate” (0–3 points). Only the two items “tingling sensations” (67%) and “numbness” (71%) were rated less often by two and one individual(s) in the range from 0 to 3 points. Out of the eight patients that described more severe (4–5 points) tingling sensations (19%) and numbness (11%), two had a mild sensory deficit in the lower extremities.

### 3.3. Clinical Findings

The clinical examination revealed mild to moderate sensory deficits affecting the lower half of the lower limbs in 7%, which partially correlated with the subjective description of numbness. Fifteen percent described numbness in the affected pain areas. In half of them, the clinical examination did not reveal any objectifiable sensory deficits. In the other half, we found a loss of sensitivity to touch, temperature perception, and pinprick discrimination. The deficits were located in the distal half of the lower legs. Vibration perception was normal in all cases. In 75% of these patients, we obtained a normal QST and the painDETECT questionnaire did not show significant signs of a neuropathic pain character.

Clinical signs of myopathy were unsteady gait and inability to squat in 20%, and 4% showed a positive Gower’s sign. Muscle weakness (≤4/5 acc. to MRC) affected hip flexors in 41%, hip extensors in 26%, knee extensors in 24%, knee flexors in 24%, arm extension in 11%, arm abduction in 9%, and finger spreaders in 6%. Furthermore, we observed atrophies (proximal: 13%, distal: 9%), scoliosis (11%), scapular winging (11%), fasciculations (7%), myotonia (8%), myoclonus and rippling (2%), ptosis (5%), and a signe des cils (5%).

### 3.4. Sensory Profiles

Compared to healthy controls, the QST profiles in patients with chronic muscle pain showed a significantly lower Z-Score for CDT, WDT, and TSL, showing hypoesthesia for both cold and warmth perception. A significantly elevated Z-Score for MPT indicated mechanical hyperalgesia, and a significantly lower Z-Score in VDT showed vibrotactile hypoesthesia. We observed significantly more paradoxical heat sensations (PHS), and the significantly elevated DMA indicated allodynia in our muscle pain cohort (Figure 2, Appendix A).

### 3.5. Carbamazepine Treatment: Duration, Dosage, Side Effects

In this study, the only indication for carbamazepine treatment was chronic muscle pain. With a mean duration of carbamazepine intake of 13.4 ± 22 months (range 0–120 months), the average daily dosage was 253.7 ± 138.3 mg (range 100–900 mg). We observed continuation in 17%, and 15% did not report any side effects. Seven percent of patients received a carbamazepine prescription but decided not to take it. Forty-four percent stopped taking carbamazepine due to side effects. The most commonly reported side effects leading to drug discontinuation were elevated liver enzymes, nausea, or fatigue (double entries possible, Table 2). Other reasons for stopping carbamazepine intake were either no improvement of muscle pain (11%) or interactions with birth control and pregnancy (2%). In 33%, no follow-up data were available, including the overall duration of medication. A detailed list of reasons for stopping carbamazepine is shown in Table 2.

### 3.6. Response to Carbamazepine

In a standardized retrospective assessment, 48% of patients reported a relevant improvement of muscle pain under therapy, while 15% did not observe any effect, and 7,5% were unsure. The average NRS before carbamazepine treatment was 6.8 ± 1.6 (range 3–10, 10 in total) while being 4.77 ± 1.99 (range 1.5–9, 10 in total) under treatment. The pain reduction (mean: 1.9 ± 1.8 points, range 0–7, 10 in total) was highly significant in this cohort (*p* < 0.001) (Figure 3a). There was no significant difference in pain reduction under carbamazepine treatment in patients with or without signs of C and Aδ nerve fiber involvement in the QST, but the pain reduction was significantly higher (*p* = 0.0359) in patients with a nociceptive painDETECT-score compared to patients with an unclear or neuropathic result (Figure 3b,c). Response to carbamazepine correlated with the initial NRS level (r = 0.52, *p* < 0.01), meaning that patients with higher NRS levels reported a greater pain reduction under treatment.

## 4. Discussion

We herein describe a representative real-life cohort examined at a German neuromuscular reference center, showing that the range of underlying conditions associated with muscle pain is broad, making it even more challenging to address this type of pain in clinical practice.

To date, there is no approved drug available specifically for the symptomatic treatment of muscle pain. Ours and other studies [40,41,42] show that muscle pain is a common and relevant symptom in many different types of neuromuscular diseases.

In our study, carbamazepine treatment significantly lowered the reported pain levels. The mechanism of action might therefore be less specific for neuropathic, but also apply for nociceptive pain. It is known that carbamazepine blocks voltage-gated sodium channels, such as Nav1.7, and nociceptors that are highly expressed in peripheral nerve endings [43]. The generated action potentials are conducted by thin and unmyelinated C and Aδ nerve fibers, which are typically damaged in neuropathic pain syndromes. What is, however, the mechanism of action in nociceptive pain? Why does carbamazepine attenuate myalgia?

Patients with myopathy show small fiber dysfunction on QST when examined over the skin of mainly foot dorm. It is possible that nociceptive fibers in the muscles themselves are also affected, in which case carbamazepine would co-treat a possible neuropathic pain component. However, this was hardly the case, as shown in Figure 3c. The mean NRS decrease was only approximately 1 pt. for the possible neuropathic myopathy group. The nociceptive myopathy subgroup showed significantly greater pain reduction, suggesting that small fiber neuropathy demonstrated by QST is not the key predictor of carbamazepine response. The mechanism of action might take place in the muscle itself, involving Nav1.4 sodium channels. Pain relief could then be explained secondarily as a result of a decrease in spontaneous muscle spasms or cramps by reduction of ectopic activation there or a more fundamental reduction of at least phasic increased muscle tone. Both mechanisms would contribute to the activation of nociceptors in muscle tissue without the need for nerve damage to be causative.

Sensory profiles can, however, vary between tissues—there is, unfortunately, no method available to directly examine the nerve fiber function in muscles. Interestingly, myopathy patients showed sensory plus signs such as pinprick hyperalgesia and dynamic mechanic allodynia as well as paradoxical heat sensations upon cooling. These measures are consistent with the concept of a possible secondary central sensitization of the nociceptive system [38] either triggered upon disturbed afferent input from deeper tissue such as the affected muscles or damaged peripheral small nociceptive nerve fibers.

Muscle pain is a complex pain syndrome that has mostly been considered nociceptive. In this work, we show that patients with nociceptive pain may still have small fiber dysfunction, which adds to the complexity of pain and might complicate therapeutic choices. In accordance with our results, other studies on nociceptive pain syndromes suggest that central sensitization might likewise play a role in non-neuropathic pain syndromes [44,45].

Despite an apparently good treatment response, 25 out of 54 patients discontinued carbamazepine treatment within six months (mean). This was mostly related to fatigue (15%), no improvement (11%), and liver enzyme elevation (7%). These side effects have previously been reported in association with carbamazepine, independent from its indication [46]. Compared to patients with epilepsy or neuropathic pain (average dosage in most cases 600 mg per day) [1], notwithstanding, the administered daily dosage was relatively low (253.7 ± 138.3 mg) in our cohort. Another interesting finding was an increase in GGT and CK levels under treatment, which did not correlate with ALT and AST levels. It should therefore be taken into consideration that patients with neuromuscular diseases might be more vulnerable to side effects.

Carbamazepine has a long-term tradition as an anti-seizure drug, and its indication was extended for neuropathic pain syndromes, including diabetic neuropathy [1] and trigeminal neuralgia [2,3]. No study has ever assessed the effects of carbamazepine on chronic muscle pain before.

In 2018, mexiletine was approved for the symptomatic treatment of non-dystrophic myotonia (paramyotonia congenita and myotonia congenita) [12] caused by mutations in chloride (*CLCN1*) and skeletal muscle voltage-gated sodium (*SCN4A*) channels [47]. Mexiletine is—like carbamazepine—a sodium channel blocker [48] previously used as an antiarrhythmic drug with high affinity for muscle sodium channels. Models suggest that mexiletine reduces muscle fiber excitability [12]. In 2020, a follow-up study proved a sufficient safety of mexiletine [49] in the aforementioned disorders, but like carbamazepine, it has never been systematically tested for chronic muscle pain in general. Contrarily, the indication is very limited and cardioarrhythmic disorders need to be ruled out and monitored under treatment.

As a potential limitation of this study, the examined cohort was heterogeneous, including various underlying conditions (Figure 1b). It is a common phenomenon that especially very rare diseases are underrepresented in clinical studies, which is why case numbers cannot meet the calculated power aims. We, therefore, refrained from conducting statistical analyses comparing our sub-cohorts but drew our conclusions based on the overall cohort. Likewise, we did not draw conclusions that would be specific for male or female patients (Appendix A). Furthermore, the assessment of muscle pain intensity before and under treatment was retrospective, and no placebo-controlled study arm was implemented in this design. The intention of this study was, however, not primarily to conduct a phase III clinical trial, but to assess the benefits and risks of this commonly used drug under real-life conditions.

Acknowledging that randomized, placebo-controlled, and double-blinded studies are necessary for high-level evidence, we admit that our retrospective study design comes with certain limitations: The time point, duration, and dosage of carbamazepine treatment were not standardized. The evaluation of treatment response was based on the subjective evaluation of pain intensity reported by patients, coming with a potential recall bias, a problem that is, however, difficult to overcome in any type of pain condition, especially in a retrospective study design. A selection bias might have arisen from the fact that we only received feedback from patients who were motivated to participate in the study.

## 5. Conclusions

We conclude that carbamazepine can be a useful medication and improve quality of life in patients with neuromuscular disorders. Due to the high rate of side effects leading to discontinuation, the expected beneficial effects of carbamazepine should be carefully weighed up against the potential risks.

## Figures and Tables

**Figure 1 brainsci-13-00123-f001:**
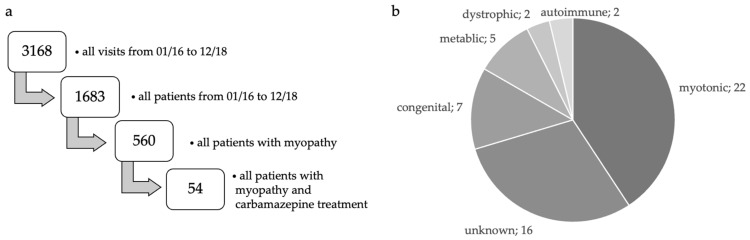
(**a**,**b**). Study collective. (**a**) Screening process and selection of our study cohort. Out of 1683 patients and 3168 visits, 54 patients fulfilled our inclusion criteria, which were adult myopathy patients (>18 years) that were symptomatically treated with carbamazepine due to chronic muscle pain. Patients with carbamazepine treatment due to other indications were not included. (**b**) Overview of all included muscle diseases. The most frequent diagnoses were myopathies of unknown origin and cramp-fasciculation syndrome, including benign forms and neuromyotonia. We also included very rare diseases such as nemaline myopathy and central core myopathy.

**Figure 2 brainsci-13-00123-f002:**
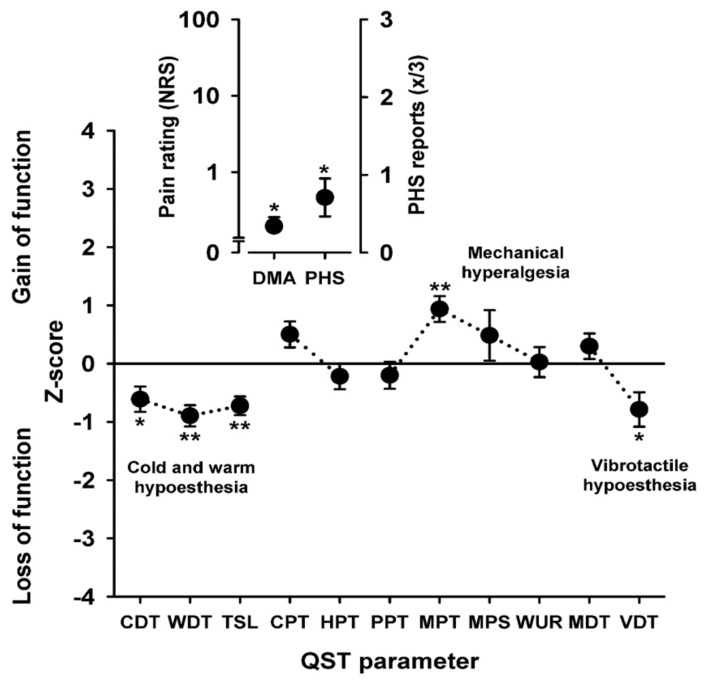
Sensory profiles. Z-score sensory profile of all patients (*n* = 24) compared with a sex-, age -, and location-matched control group (*n* = 24). The profile shows loss of function for non-painful thermal stimuli (CDT, WDT, TSL) corresponding to small fiber dysfunction. The presence of mechanical hyperalgesia to pinprick (MPT) and dynamic mechanic allodynia (DMA) points towards possible secondary central sensitization of the nociceptive system. The vibrotactile hypoesthesia (VDT) is consistent with large fiber involvement in this patient group. The increased number of paradoxical heat sensations (PHS) might be attributed to peripherally or centrally disturbed thermal discrimination. Stars denote the level of significance with * *p* < 0.05; ** *p* < 0.01 (one-way ANOVA).

**Figure 3 brainsci-13-00123-f003:**
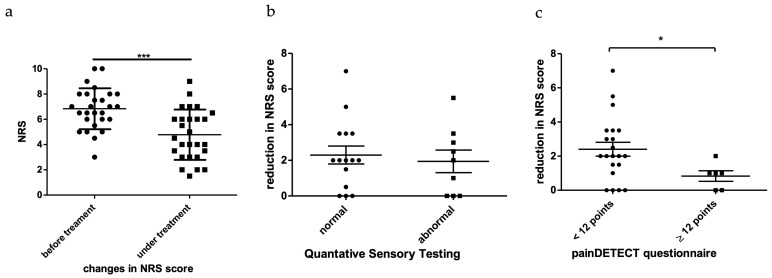
(**a**–**c**). NRS evaluation. (**a**) Response to therapy with carbamazepine. Patients were asked to score their pain levels before and under carbamazepine treatment. The NRS level was significantly lower under treatment. (**b**) NRS reduction did not correlate with QST results. The y-axis depicts the change in NRS before and under treatment. There was no significant difference in pain reduction between patients with normal and abnormal QST results. (**c**) NRS reduction correlated with painDETECT-scores. The y-axis depicts the change in NRS before and after treatment. Pain reduction was significantly greater in patients with a normal painDETECT questionnaire (*p* < 0.05) than in patients, whose questionnaire showed unclear results or results indicative of neuropathic pain. Stars denote the level of significance with * *p* < 0.05; *** *p* < 0.001.

**Table 1 brainsci-13-00123-t001:** Patient overview and demographics *.

			Unknown	Metabolic	Congenital	Myotonic	Dystrophic	Autoimmune
**sex**	m:f (age in years)		9:7 (49.5:47.4)	4:1 (50.65:25)	5:2 (48.13:56)	22:0 (55.43)	1:1 (80:60)	1:1 (54:46)
**general** **information**	AOO (years)		33.19	24.3	33	35.99	52	45
	duration (years)		15.45	19.6	20.25	19.32	13	4
**first** **symptoms**	myalgia		16	5	6	18		1
	muscle weakness		3	2	1	4	2	
	muscle cramps		5	1	2	12		1
	fasciculations		3			12		
	muscle stiffness		1			3		1
	rhabdomyolysis		1					
	elevated CK levels		1		3			
	arthralgia					1		
**current** **symptoms**	myalgia		16	5	6	18		1
	muscle weakness		6	2	2	6	2	
	muscle cramps		6	1	1	11		1
	fasciculations		3			8		
	muscle stiffness		2			1		
	elevated CK levels				1			
	hypermobile joints		1					
**signs of** **myopathy**	unsteady gait		5		1	2	2	1
	Gower’s sign			1	1			
	inability to squat		6	1		4		
	muscle atrophies		4		1	2	2	1
	scapular winging		4				2	
	scoliosis		4		1	1		
	fasciculations				1	3		
	myotonia				1	3		
	myoclonus				1			
	rippling			1				
	weakness				1			
	ptosis		1			1	1	
	signe des cils		1				2	
**deep tendon** **reflexes**	normal		14	4	7	19		1
	reduced		2	1	1	3	2	1
**MRC [x/5]**	proximal weakness	yes	8	3	3	4	2	2
		no	8	2	4	18		
	distal weakness	yes	6	3	2	3	2	
		no	10	2	5	19		2
**NRS**	before treatment		6.19	9	7	6.19	no data	7
	under treatment		5	7.5	4.33	4	no data	6.5
**carbamazepine**	dosage (mg)		200	200	200	401.39	200	200
	duration (months)		4.55	12	13.83	45.6	8	49
	improvement	yes	7	5	2	11		1
		not sure	3		2	2	1	
		no	3		2	1	1	
**QST**	normal		7		2	4	no data	1
	abnormal		3	1	2	3	no data	

*: Based on etiology, the table is divided into six groups: unknown, metabolic, congenital, myotonic, dystrophic, and autoimmune. Each subgroup contains different muscle diseases. Confirmed myopathies with, however, unknown causes are summarized in the “unknown” category (*n* = 16). The metabolic subgroup (*n* = 5) consists of metabolic myopathy, mitochondrial myopathy, and phosphorylase deficiency (Mc Ardle disease). Myofibrillar myopathy, central core disease, centronuclear myopathy, and nemaline rod myopathy were summed up as congenital myopathies (*n* = 7). In this cohort, FSHD was the only muscle dystrophy reported with relevant muscle pain (*n* = 2). Neuromyotonia, myotonia congenita type Becker, paramyotonia congenita, and PROMM form the category “myotonic” (*n* = 22). As autoimmune (*n* = 2) counted autoimmune necrotizing myopathy and myositic overlap syndrome. A more detailed table can be found in the supplementary material (Appendix A).

**Table 2 brainsci-13-00123-t002:** Reported side effects *.

Adverse Side Effects Observed under Carbamazepine	Occurrence	Carbamazepine Stopped
not specified	18	no data
none	8	0
fatigue	8	7
no improvement	6	6
liver enzyme elevation	4	4
diarrhea	2	2
hospitalization due to skin condition	2	2
family planning	1	1
hypersomnia	1	1
eczema	1	1
hypersensitivity to sunlight	1	1
carbamazepine-induced LE	1	1
nausea	1	1
mood swings	1	1
depression	1	1
intolerance	1	1
vertigo	1	1
stomach ache	1	1
total	59	32

*: The table shows an overview of all reported side effects (*n* = 59). Patients were allowed to name more than one side effect (*n* = 6). Roughly, one quarter was not able to specify the experienced side effects (*n* = 18), and 4 patients decided to not take carbamazepine after receiving the first prescription. Overall, there were 8 patients that did not report any side effects.

## Data Availability

Original data are available and can be obtained from the corresponding author by reasonable request.

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
