# Peer review of "Carbamazepine for Chronic Muscle Pain: A Retrospective Assessment of Indications, Side Effects, and Treatment Response"

_brainsci, 2023, doi:10.3390/brainsci13010123_

Round 1

Reviewer 1 Report

Here, Dyong et al. applied a retrospective evaluation on effects of voltage-gated sodium channel blocker Carbamazepine on chronic muscle pain patients. In the manuscript, the authors combined methods of painDETECT questionnaire and quantitive sensory testing and assessed sensory responses, Carbamazepine treatment effect as well as side effects of the patients. This manuscript provided proper data that supported the major arguments. However, there are still some issues within the paper that needs further changes, including: 

1. Starting from the line 64, the last paragraph of the introduction showed no relevance to all the other ones , which looked like an instruction for drafting a proper introductory texts. This whole paragraph needs to be rewritten, replaced or removed from the introduction. 

2. In the line 159, it was introduced that 42 males and 12 females subjects were included in this study. However, the whole dataset may demonstrate gender bias towards the male. In this sense, including more female subjects into the study, or making a sex-specific analysis of all the data may give a clear answer whether Carbamazepine has a differential effect on male and females. 

3. A typo needs to be changed in the line 160, "46 suffered from hereditary..." should be "46% suffered from hereditary...". 

4. In the line 215 and many other sites, representation by "4.2+/-2.7/10" is confusing. "4.2+/-2.7 (10 in total)" may work better. 

5. In Figure 2, a cohort of quantitative sensory testing was analyzed with Z-score instead of the actual readout. Considering all the Z-score values showed are relatively small (between -1 to 1), it would be necessary to include the actual sensory test readout without the Z-score analysis. 

6. In Figure 3a, individual data points needs to be shown in the panel. 

In overall, the manuscript should be considered with a major revision before reevaluation. 

Author Response

Dear Editor-in-Chief, dear Reviewer 1,

Please find our revised manuscript “Carbamazepine for chronic muscle pain: a retrospective assessment of indications, side-effects, and treatment response” by Dyong et al., which we would like to re-submit to Brain Sciences as an Original Article.

In the following passages, we will address the reviewers’ comments point by point.

Point 1: Starting from the line 64, the last paragraph of the introduction showed no relevance to all the other ones , which looked like an instruction for drafting a proper introductory texts. This whole paragraph needs to be rewritten, replaced or removed from the introduction.

Response 1:      Thank you for pointin this out. We deleted it accordingly (see line 61).

Point 2:  In the line 159, it was introduced that 42 males and 12 females subjects were included in this study. However, the whole dataset may demonstrate gender bias towards the male. In this sense, including more female subjects into the study, or making a sex-specific analysis of all the data may give a clear answer whether Carbamazepine has a differential effect on male and females. 

Response 2:      Thank you for this valuable comment. We agree that gender differences might play a considerable role in the context of pain and pain therapy. Considering that this was a retrospecitve assessment in patients that had received Carbamazpine for chronic muscle pain prior to enrollment, however, the design of this study did not allow for an equally pooled distribution in the first place.  Statistically, there actually was a significant difference between NRS score changes in males and females, however,  taking into account your comment, unequal group numbers (male n= 17 and female n= 7) and durations of medication intake (females might have ended or paused carbazepine treatment due to interactions with contraception or potential teratogeneity) made us refrain from interpreting this finding. We added a table to the supplementary material, showing our original NRS data for both sexes. We also discussed our results as follows:  “We compared the change in NRS (numeric rating scale) scores in male (n= 17) and female (n= 7) patients separately. The NRS results before and after treatment were not significantly different (male to female before treatment: p= 0.3024, male to female after treatment: p= 0.335). However, the change in NRS scores was significantly different (p= 0.0245). From this retrospectively assessed real-life cohort, it is difficult to extrapolate a clinically relevant sex difference in pain reduction. Further studies with larger cohorts and a prospective, double-blind study design are needed to further explore this effect.” (see supplementary result S2). In the discussion of the main manuscript, we added the sentence: “Likewise, we refrained from drawing conclusions that would be specific for male or female patients (supplementary results S2).” (Ll. 433/434).

Point 3: A typo needs to be changed in the line 160, "46 suffered from hereditary..." should be "46% suffered from hereditary...". 

Response 3:      We changed it accordingly. (see line 181)

Point 4:             In the line 215 and many other sites, representation by "4.2+/-2.7/10" is confusing. "4.2+/-2.7 (10 in total)" may work better. 

Response 4:      We changed it accordingly for better visibility and clarification. (see lines 242, 260, 334, 335 and 336)

Point 5: In Figure 2, a cohort of quantitative sensory testing was analyzed with Z-score instead of the actual readout. Considering all the Z-score values showed are relatively small (between -1 to 1), it would be necessary to include the actual sensory test readout without the Z-score analysis. 

Response 5:      We appreciate your comment regarding the original QST results and agree that showing this data is important. That is why we included an according table in the supplementary material. In the main manuscript, we prefered to show z values, since this is the already established way of illustrurating nerve fiber gain- and loss-of-function, depicted by their specific QST-subitems. For comparison, please see DOI: https://doi.org/10.1016/j.pain.2006.01.041; https://doi.org/10.1016/j.ejpain.2005.02.003.

Point 6:             In Figure 3a, individual data points need to be shown in the panel. 

Response 6:      Thank you, this totally makes sense to us. We changed it accordingly. (see line 342)

We hope that our manuscript will find acceptance for publication in Brain Sciences.

Sincerely yours,

Maike F. Dohrn

Reviewer 2 Report

The authors have developed an interesting retrospective study on the use of Carbamazepine to treat a variety of myopathies.

However, I suggest several clarifications and/or modifications that will in my opinion improve the quality of their manuscript:

1. Please, define the objectives well at the end of the introduction, and delete the remaining text.

2. Please, Remove excess text at the end of the description of the statistical analysis.

3. It is difficult to follow the values obtained for each of the variables in Table 1...Please improve its visibility.

4. During the reading of the manuscript, it is frequent to find the numbers written as well as in Arabic...please, homogenize.

5. Given the nature of their work and their findings on signs and symptoms of altered central nervous system processing and types of pain, I recommend that the authors enrich the discussion by commenting on other musculoskeletal disorders and the alteration of pain modulation mechanisms, as well as the coexistence of muscular problems in processes that are often considered articular: DOI: 10.3390/app11041895; DOI: 10.3390/jcm9082561

6.Could the authors add a "Conclusions" section?

Author Response

Dear Editor-in-Chief, dear Reviewer 2, 

 Please find our revised manuscript “Carbamazepine for chronic muscle pain: a retrospective assessment of indications, side-effects, and treatment response” by Dyong et al., which we would like to re-submit to Brain Sciences as an Original Article.  

In the following passages, we will address the reviewer’s comments point by point. 

Point 1: Please, define the objectives well at the end of the introduction, and delete the remaining text. 

Response 1: We deleted the according parts in the introduction (see line 64). 

Point 2: Please, Remove excess text at the end of the description of the statistical analysis. 

Response 2: Thank you for making us aware. We deleted the according part (see line 160). 

Point 3: It is difficult to follow the values obtained for each of the variables in Table 1...Please improve its visibility. 

Response 3: Thank you for the valuable feedback. We revised table 1 for better visualization of variables (see line 206). 

Point 4: During the reading of the manuscript, it is frequent to find the numbers written as well as in Arabic...please, homogenize. 

 Response 4: Thank you for the feedback. Since we did not find specific instructions for spelling out numbers in accordance with the journal style, we decided to abide with the default way of using numbers in English. As far as we know, this means to spell out numbers at the beginning of a sentence as well as numbers from one to twelve, whereas numbers 13 and above would not be spelled out. Beyond these standards, we are happy to change the way of enumerating: Please advise if there is a speficic style that the reviewer or the editor would prefer. 

Point 5: Given the nature of their work and their findings on signs and symptoms of altered central nervous system processing and types of pain, I recommend that the authors enrich the discussion by commenting on other musculoskeletal disorders and the alteration of pain modulation mechanisms, as well as the coexistence of muscular problems in processes that are often considered articular: DOI: 10.3390/app11041895; DOI: 10.3390/jcm9082561 

Response 5: We added the following section to the discussion: “Muscle pain is a complex pain syndrome that has mostly been considered nociceptive. In this work, we show that patients with nociceptive pain may still have small fiber dysfunction, which adds to the complexity of pain and might complicate therapeutic choices. In accordance with our results, other studies on nociceptive pain syndromes suggest that central sensitization might likewise play a role in non-neuropathic pain syndromes[44, 45].”(see line 395ff). References 44 and 45 are the ones suggested by the reviewer. 

Point 6: Could the authors add a "Conclusions" section? 

Response 6: We added the following section: “We conclude that carbamazepine can be a useful medication and improve quality of life in patients with neuromuscular disorders. Due to the high rate of side effects leading to discontinuation, the expected beneficial effects of carbamazepine should be carefully outweighed with the potential risks.” (see line 460ff) 

We hope that our manuscript will find acceptance for publication in Brain Sciences. 

 Sincerely yours, 

 Maike F. Dohrn 

Round 2

Reviewer 1 Report

The authors have addressed the majority of the major questions within the comments. The manuscript should be considered accepted with present form. 

Reviewer 2 Report

The authors have responded to all my requests.

In addition, their manuscript has been substantially improved, so I recommend the current version for publication.

Congratulations